# Novel Structure and Thermal Design and Analysis for CubeSats in Formation Flying

**Yeon-Kyu Park, Geuk-Nam Kim** 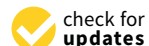 **and Sang-Young Park ***

Astrodynamics and Control Laboratory, Department of Astronomy, Yonsei University, Seoul 03722, Korea;
yeonqpark@gmail.com (Y.-K.P.); south920528@gmail.com (G.-N.K.)
**\*** Correspondence: spark624@yonsei.ac.kr

**Abstract:** The CANYVAL-C (CubeSat Astronomy by NASA and Yonsei using a virtual telescope alignment for coronagraph) is a space science demonstration mission that involves taking several images of the solar corona with two CubeSats—1U CubeSat (Timon) and 2U CubeSat (Pumbaa)—in formation flying. In this study, we developed and evaluated structural and thermal designs of the CubeSats Timon and Pumbaa through finite element analyses, considering the nonlinearity effects of the nylon wire of the deployable solar panels installed in Pumbaa. On-orbit thermal analyses were performed with an accurate analytical model for a visible camera on Timon and a micro propulsion system on Pumbaa, which has a narrow operating temperature range. Finally, the analytical models were correlated for enhancing the reliability of the numerical analysis. The test results indicated that the CubeSats are structurally safe with respect to the launch environment and can activate each component under the space thermal environment. The natural frequency of the nylon wire for the deployable solar panels was found to increase significantly as the wire was tightened strongly. The conditions of the thermal vacuum and cycling testing were implemented in the thermal analytical model, which reduced the differences between the analysis and testing.

**Keywords:** CubeSat; finite element analysis; ground validation; analytical model correlation

## 1. Introduction

CubeSats are cheaper and faster than conventional satellites for space exploration owing to the standardized configurations of CubeSats bus and miniaturization of payload. CubeSat platforms have been applied for multi-satellite missions, such as constellation missions and formation flying missions [1]. The PlanetScope of PlanetLabs provides high-resolution optical images of the entire planet with a short visit time and a constellation of hundreds of 3U CubeSats [2]. The SNIPE mission by KASI aims to study the near-Earth space environment through four formation-flying 6U CubeSats [3]. To meet the growing demand for multi-CubeSat missions, mass production is required, which in turn requires efficient analytical models and ground validations.

Researchers at the Yonsei University have developed two CubeSats, named Timon (1U CubeSat) and Pumbaa (2U CubeSat), for the CANYVAL-C (CubeSat Astronomy by NASA and Yonsei using Virtual Telescope Alignment for Coronagraph) mission. The aim of the CANYVAL-C mission is to photograph the solar corona using precise formation flying technologies. The technical objective of this mission is the implementation and verification of the relative guidance, navigation, and control (GNC) technology, with inter-satellite link technology. Figure 1 illustrates the concept of the CANYVAL-C mission [4]. To obtain the solar corona images, Pumbaa and Timon will be aligned with respect to the sun, resulting in an artificial eclipse.

Timon and Pumbaa have similar bus architecture. Timon is equipped with a visible camera, battery, power control and distribution unit (PCDU), on-board computer (OBC), magnetic torquer (MTQ), and deployable UHF antenna. In Timon, electrical power is

generated through body-mounted solar panels on each axis, and the MTQs on each axis are air-coil-type, which save internal space and implement attitude control. Additionally, Pumbaa is equipped with a propulsion system (PPS), reaction wheel assembly (RWA), deployable solar panel (DSP), and occulter made of a polyimide film and four tape springs that blocks sunlight. The CubeSats have been constructed following the proto-flight model (PFM) development philosophy, i.e., environment testing was conducted only at the integrated systems level. The launch environment conditions, such as random vibrations, acceleration, shocks, sound, and resonance can cause structural damage to each CubeSat, determined in terms of deformation and stress [5–7]. To evaluate the structural safety of the CubeSats, finite element analyses (FEAs) were conducted using the NX10.0 and NASTRAN solvers by Siemens. Vibration testing was conducted in a simulated launch environment to validate the structural design. In the space thermal environment, the major external heat sources are solar radiation, earth albedo, and infrared [8]. Because heat transfer does not occur through convection owing to the vacuum environment in orbit, heat transfer only occurs through conduction and radiation. Because of the orbital motion, the temperature changes significantly, and these limited heat sources induce damage to the mechanical properties of the CubeSats [9–12]. The on-orbit thermal analysis of the CubeSats was also executed through the FEAs, using the NX10.0 and Space Systems Thermal solver by Siemens. Based on the temperature profiles from the thermal analysis, thermal vacuum and cycling tests (TVCTs) were carried out simultaneously with the two CubeSats in a thermal vacuum chamber.

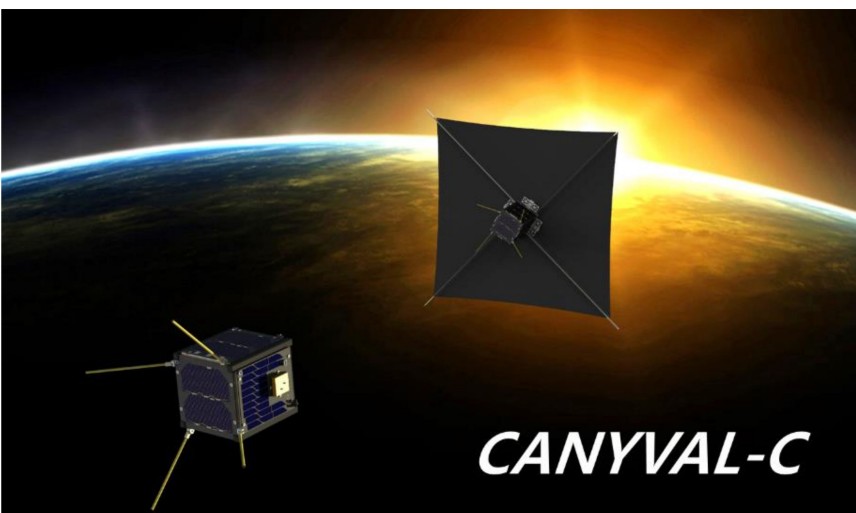

**Figure 1.** Concept of CANYAL-C mission [4]. The CubeSat on the left is Timon (1U CubeSat) and that on the right, with a deployed occulter, is Pumbaa (2U CubeSat).

In this study, we present and validate the structural and thermal design of the CubeSats for the CANYVAL-C mission through FEAs and environmental tests. First, we investigate the nonlinearity of the DSP by implementing an analytical model of a nylon wire for accurate numerical analysis. Pumbaa is equipped with the DSP to enhance electrical energy generation. The hold and release mechanism (HRM) is the part of the DSP that holds the panels to minimize the size of the exterior configuration during the launch phase and releases the panels during orbit in order to charge the battery. Generally, the 1st mode natural frequency ($f_0$) of the DSP with HRM varies during the vibration test owing to irregular design and backlash of the torsional hinges [13]. Therefore, to guarantee structural stability without many physical verifications, the characteristics of the DSP and HRM with the tightened nylon wire were duplicated in the structural analytical model. Second, we implemented constraints by coupling multiple CubeSats in a single deployer. Previous studies only addressed the constraints on the CubeSat motions with a deployer, which is suitable for a 3U CubeSat in a single deployer [14]. However, for the CANYVAL-C

mission, two CubeSats are contained in a single deployer, which induces the coupling conditions between each CubeSat. The coupling conditions between the CubeSats affect the magnitude of the response to the vibration and make $f_0$ lower than in the case with a single CubeSat in a single deployer. The structural designs of each CubeSat were developed considering the relevant coupling effects with respect to the launch environment, including the configuration of the DSP and the HRM, and arrangements of each component in the frame. Additionally, the thermal analytical models were correlated by comparing the results of the TVCTs and numerical analyses. The heat transfer between the CubeSats and the chamber were simulated in the numerical analysis. The camera on Timon, the micro-PPS on Pumbaa, and the Li-Polymer (LiPo) battery on both have very narrow operating temperature ranges. Thus, they require active or passive thermal control with accurate predictions of the temperature change profiles of each component. In particular, the LiPo battery and PPS are equipped with built-in heaters for active thermal control. The required duty rate of the heaters was determined using a thermal analytical model for the power budget analysis of the systems. Finally, based on the correlated analytical model, the structural characteristics of the two CubeSats with different sizes, which cause temperature differences, were studied. In previous studies, TVCTs for most CubeSat missions was executed only for a single CubeSat. However, in this study, TVCTs were conducted for multiple CubeSats simultaneously, which is important for understanding the cause of the temperature difference between the two CubeSats to ensure a reliable correlation scheme.

The rest of this paper is organized as follows. The detailed design and analytical model of the two CubeSats are described in Section 2. The pre-processing with the constraints and the results of the structural analysis and on-orbit thermal analysis are presented in Section 3. The ground validations by launch and space environment testing are presented in Section 4. The correlations of the analytical models based on the results of environmental tests are reported in Section 5. Finally, concluding remarks are provided in Section 6.

## 2. CubeSat Design

### 2.1. Structural Design

The CubeSats are composed mainly of Al 6061, SS304, FR-4, and copper. In particular, Al 6061 is the main material for the chassis owing to its mechanical rigidity and chemical stability. The chassis surfaces are anodized to prevent cold welding in the space environment [15]. To achieve precise attitude and orbit maneuvers, the center of mass is located within 10 mm of the geometric center in each axis. Figure 2 shows the configuration of the CubeSats Timon and Pumbaa.

The payload of Timon is a visible camera. Timon generates electrical power using body-mounted solar panels on each axis. The MTQs on each axis are air-coil types, which require less internal space and implement attitude control. The configuration of Pumbaa is similar to that of Timon. However, Pumbaa is specially equipped with a PPS, RWA, DSP, and occulter, as shown in Figure 2. To prevent physical interference with the deployer, the size of the extrusions including the visible camera, global navigation satellite system (GNSS) antenna, and DSP is less than 10 mm.

### 2.2. Analytical Model Design

Detailed analytical models require high computational power and long analysis time. These parameters can be reduced by removing unnecessary parts and the parts that do not affect physical phenomena. Figure 3 shows the detailed computer-aided design (CAD) model and structural and thermal analytical models of the CubeSats considered in this study. For the structural analytical model, neglectable parts such as connectors, chamfers, and filets are removed. However, fastening configurations, such as threaded screws and hinges, are required to preserve the physically coupled characteristics. The thermal analytical model was simpler than the structural analytical model as it omitted configurations with little heat exchange. In particular, the thermal analytical models only include major heating parts, such as processors, resistors on the HRM, and heaters.

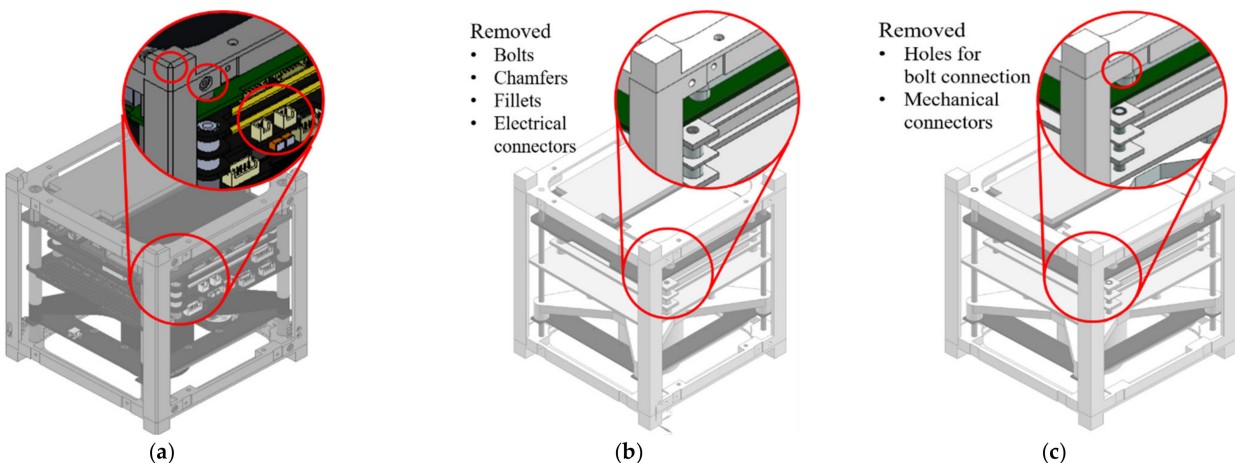

**Figure 2.** CubeSat hardware configuration and coordinate system: (**a**) 1U CubeSat, Timon; (**b**) 2U CubeSat, Pumbaa.

**Figure 3.** CAD and analytical models: (**a**) Detailed CAD model and (**b**) structural and (**c**) thermal analytical models of Timon.

### 2.3. Launch Environment

The CubeSats were designed according to the PFM development philosophy. The launch vehicle is a Soyuz-2.1a (Progress Inc., Samara, Russia). The structural safety of the

CubeSats under the launch environment was evaluated through a structural analysis of a modal survey, quasi-static acceleration, random vibration, and shock to the qualification level provided by the innovative space logistics [16].

The analysis results were evaluated in terms of the $f_0$, maximum deformation, design safety factor (DSF), and margin of safety (MoS). The $f_0$ set to more than 50 Hz for each axis to avoid resonance with the launch vehicle and deployer. The maximum deformation did not cause physical interference among the components. The recommended minimum DSF depends on the type of verification approach, structure geometry, and materials [17]. Given complex structures with various components, this study adopts a value of 2.0 as the DSF, which is the highest DSF value, representing the worst-case scenario during the launch phase. The MoS was simply calculated as the ratio of the allowable stress to the maximum stress; when the MoS is greater than zero, the CubeSats is considered to be safe under the launch environment.

Given the PFM philosophy, which does not include a structure-thermal model (STM) and qualification model (QM), the shock test was omitted and only shock analysis was conducted. Although the shock profile for the numerical analysis was applied with a safety factor of 1.5, the MoS was calculated with a safety factor of 2.0. According to the numerical analysis conducted without the shock test, the structure is considered safe from the shock environment.

Although the acceleration of the launch vehicle causes dynamic loads, quasi-static analyses are generally executed in structural analyses for simplicity [18]. In this study, the quasi-static acceleration for the qualification level was set to 10G in the axial direction and 9G in the lateral direction. As the random vibration caused mainly by noise and air friction is a non-deterministic vibration, structural responses with respect to an input vibration can be handled through a stochastic approach. The random vibration characteristic can be expressed in term of the acceleration spectral density (ASD), representing the power spectral density as accelerations. The characteristics of the launch vehicle, Soyuz-2.1a, are expressed in terms of the ASD levels, as shown in Table 1, ranging from 20 to 2000 Hz. The shock characteristic is expressed in terms of amplitude values in the frequency range of 100–5000 Hz.

**Table 1.** Soyuz-2.1a launch environment characteristics and random vibration test profiles.

| Frequency (Hz) | Launch Environment | | Random Vibration Test Profile | | |
| --- | --- | --- | --- | --- | --- |
| | Random Vibration | Shock | First Stage | Second Stage | Upper Stage |
| | Acceleration Spectral Density (g²/Hz) | Amplitude (g) | Acceleration Spectral Density (g²/Hz) | | |
| 20–100 | 0.01 | N/A | 0.02 | 0.02 | 0.004 |
| 100–200 | 0.01–0.025 | 30–60 | 0.02–0.05 | 0.02 | 0.004 |
| 200–500 | 0.025 | 60–255 | 0.05 | 0.02–0.008 | 0.004 |
| 500–1000 | 0.025–0.013 | 255–750 | 0.05–0.025 | 0.008–0.004 | 0.004 |
| 1000–2000 | 0.013–0.006 | 750 | 0.025–0.013 | 0.004–0.002 | 0.004–0.002 |
| 2000–5000 | N/A | 750–1500 | N/A | N/A | N/A |
| Acceleration ($G_{RMS}$) | N/A | N/A | 7.42 | 3.58 | 2.59 |
| Duration (s/axis) | N/A | N/A | 60.0 | 240.0 | 437.5 |

## 3. Numerical Analysis by Finite Element Analysis (FEA)

### 3.1. Structural Analysis

#### 3.1.1. Pre-Processing

To simplify the FEA structural analytical model, fastening conditions were established using screws, nuts, and thread parts. Gluing conditions were applied to epoxy-fixed parts. The mesh configuration was generated as a 3-D tetrahedral type. The numbers of elements and nodes were 530,295 and 1,129,293, respectively.

Figure 4 shows the configurations of the simplified CAD model, meshed model, and structural analytical model of the two CubeSats contained in the deployer. Figure 5a shows the degrees of freedom (DOFs) constraint on the rotating parts and the contact conditions between the CubeSats contained in the deployer; note that the X- and Y-axis motions of the CubeSats are strongly bounded. Additionally, the constraints for the rotational motion were applied to the hinges of the UHF antenna, DSP, and occulter. In contrast, the separation spring of the deployer allows translational motion in the Z-axis direction.

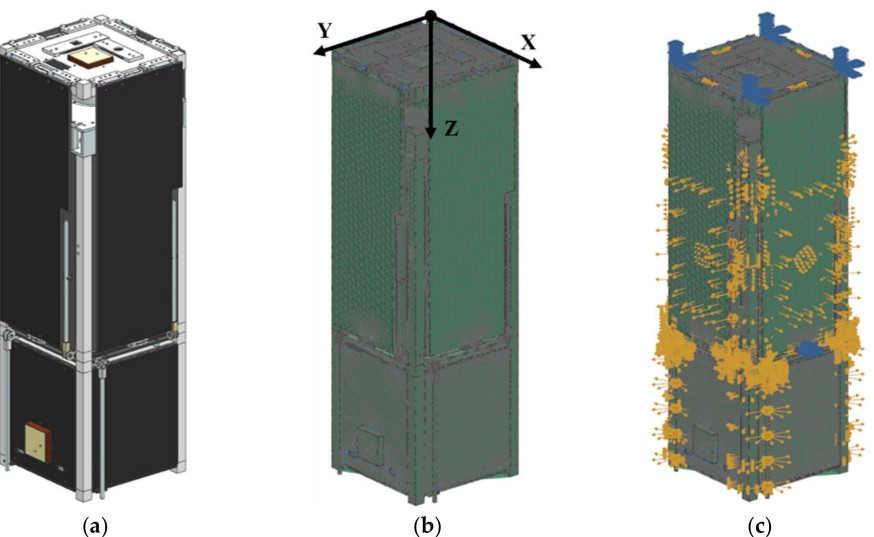

(a)                                                      (b)                                                      (c)

**Figure 4.** Models for the analyses: (**a**) Simplified CAD models of two CubeSats contained in the deployer; (**b**) 3-D tetrahedral type model; (**c**) Analytical model with contact conditions and DOF conditions.

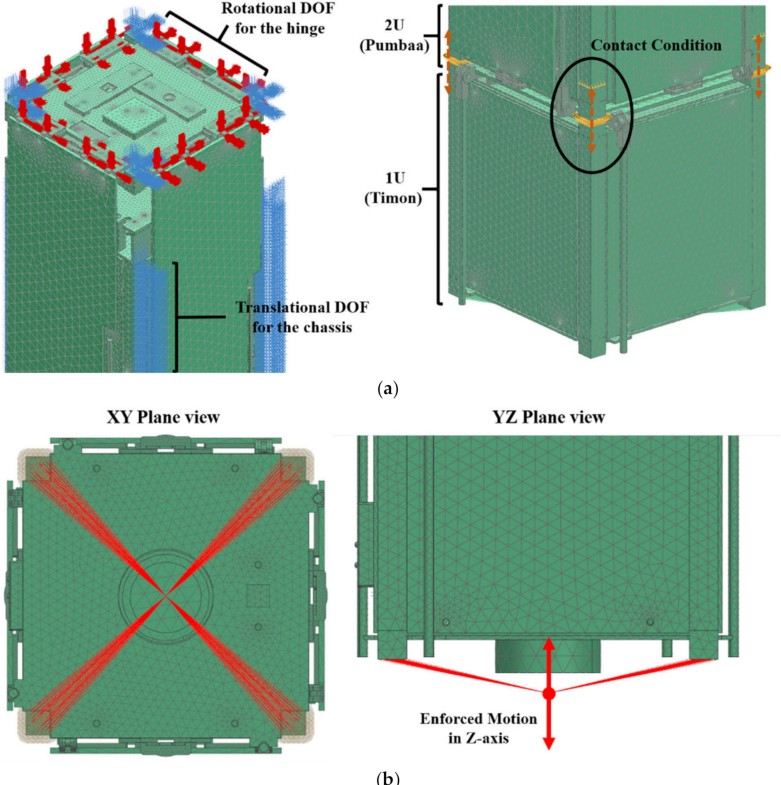

**Figure 5.** Analysis condition settings: (**a**) DOF constraints and contact conditions; (**b**) XY Plane view and YZ plane view of enforced motion for Z-axis modal analysis.

### 3.1.2. Modal Analysis

For the modal analysis, enforced motion was applied in each axis, and the displacement, speed and acceleration were set by the enforced motion to calculate the response in the time or frequency domain [19]. Enforced motion was applied to a single node that was in contact with the end of the four rails of the chassis. Figure 5b shows an example of the enforced motion along the Z-axis.

The aforementioned constraints make $f_0$ lower compared to that in the single CubeSat case. To increase $f_0$, the thickness of the DSP, which is the component most vulnerable to vibration, was increased. Figure 6 shows that $f_0$ shifts with changes in the thickness of the DSP. It can be seen that the DSP with a 2.0 mm thickness, selected as the main PCB, is much stiffer than that with a 1.2 mm thickness and that with a 1.6 mm thickness. $f_0$ in the axial and lateral directions is 120.57 and 120.55 Hz, respectively, which are greater than the stiffness requirement of 50 Hz. According to the analysis results, a thicker DSP is safer with respect to vibration. However, given the weight constraints and physical interference with the deployer, a DSP thickness of 2.0 mm is suitable for the CubeSat mission.

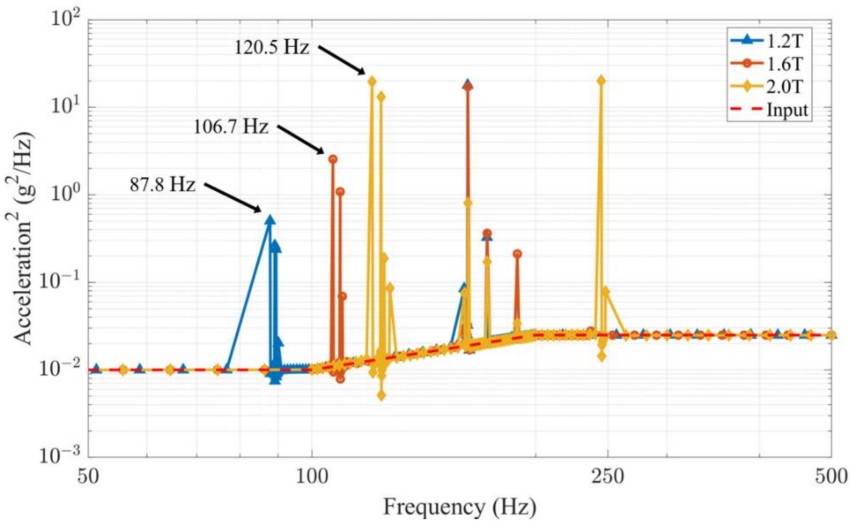

**Figure 6.** Natural frequencies ($f_0$) along the +Y-axis of the DSP in the axial direction with different DSP thicknesses.

### 3.1.3. Quasi-Static Acceleration Analysis

For the quasi-static acceleration analysis, the maximum deformation can cause physical interferences among the components, and the stress caused by the acceleration loads is calculated as the von Mises stress. Table 2 summarizes the calculated values of the maximum deformation and maximum stress, their locations, and the MoS. Figure 7 shows the results of the quasi-static acceleration analysis. It can be seen that the maximum deformation occurs on the tip of the UHF antenna in the X-axis direction, and the maximum stress occurs on the occulter's panel in the Y-axis direction. The MoS is greater than zero and there is no physical interference, indicating the safety of the CubeSats in the acceleration that occurs during the launch.

**Table 2.** Quasi-static acceleration analysis results.

| Axis | Maximum Deformation | | Maximum Stress | | | |
|---|---|---|---|---|---|---|
| | Location | Deformation (mm) | Location | Material | Stress (MPa) | MoS |
| X | 2U UHF antenna | 0.24 | 2U Payload | Al 6061 | 46.20 | 1.99 |
| Y | 2U UHF antenna | 0.22 | 2U Payload | Al 6061 | 47.40 | 1.91 |
| Z | 1U OBC spacer | 0.04 | 1U Board supporting rod | SS 304 | 11.40 | 11.1 |

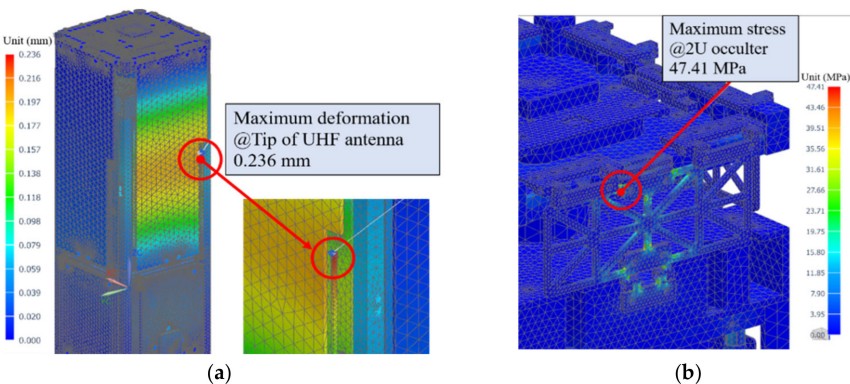

**Figure 7.** Quasi-static acceleration analysis results: (**a**) maximum deformation in X-axis; (**b**) maximum stress in Y-axis.

### 3.1.4. Random Vibration Analysis

A random vibration analysis was conducted based on the results of the modal analysis. The structural responses were predicted without changes in the constraints and contact conditions from the quasi-static acceleration analysis. Additionally, the viscous damping and hysteretic damping were set to 2%. The peak amplitude against the structural response indicated that a natural frequency existed. The lowest natural frequency was predicted to be 123.9 Hz on Pumbaa's X-axis DSP, which is consistent with the modal analysis results.

As shown in Figure 8a, the maximum stress occurs on the Pumbaa board supporting rod by random vibration along the Y-axis, and the MoS is 2.71. Table 3 summarizes the results of the random vibration analysis for each axis. From the table, it can be seen that the MoS is always greater than zero, indicating a structural stiffness sufficient for supporting the random vibrations in the launch environment.

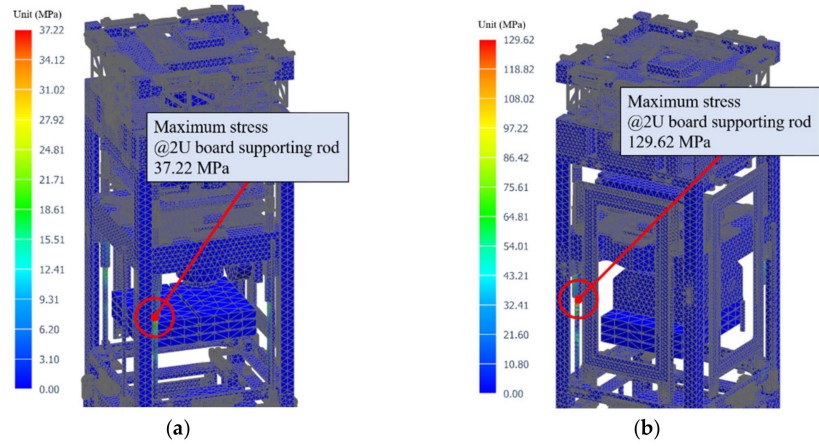

**Figure 8.** Random vibration and shock analysis results: (**a**) random vibration (maximum stress in Y-axis); (**b**) shock (maximum stress in X-axis).

**Table 3.** Maximum stress of random vibration and shock analysis.

| Axis | Random Vibration Analysis | | | | Shock Analysis | | | |
|------|----------|----------|-------------|------|----------|----------|-------------|------|
| | Location | Material | Stress (MPa) | MoS | Location | Material | Stress (MPa) | MoS |
| X | 2U Board supporting rod | SS 304 | 34.20 | 3.04 | 2U Board supporting rod | SS 304 | 129.60 | 0.07 |
| Y | 2U Board supporting rod | SS 304 | 37.22 | 2.71 | 2U Deployment device | Al 6061 | 122.20 | 0.13 |
| Z | 2U Solar array | FR-4 | 1.522 | 78.4 | 2U Hinge | Al 6061 | 7.54 | 17.30 |

### 3.1.5. Shock Analysis

The responses to shock were analyzed with the same constraints and contact conditions considered in the modal analysis. From Figure 8b, it can be seen that the maximum stress by the shock occurs on the Pumbaa's board supporting rod in the X-axis and its value is 129.62 MPa with an MoS of 0.065. The largest stress due to shock is displayed in the X-axis. Table 3 summarizes the shock analysis results. The MoS in each axis is greater than zero, indicating that the CubeSats will be safe in the shock environment during the launch phase.

### 3.2. Thermal Analysis

#### 3.2.1. Space Environment

The objective of the thermal control subsystem (TCS) is to maintain an operational temperature range of 10 °C. The mission orbit is a sun-synchronous orbit with an altitude of 500 km and a local time on the ascending node of 11:00:00.000. Figure 9 shows the orientation of the CubeSats during the imaging of the solar corona, referred to as the mission mode, and during normal operation, referred to as the normal mode, respectively. The normal mode accounts for the majority of the mission lifetime. Timon and Pumbaa orient toward the sun and perform slow slewing to maximize electrical power generation. The mission mode that photographs images of the solar corona lasts for less than three orbital periods. During the mission mode, the visible camera of Timon and the DSP of Pumbaa orient toward the sun. Moreover, most of the components on both CubeSats are activated in this mode, which results in maximum internal heat generation. An on-orbit thermal analysis was conducted for the hottest and coldest cases defined by the solar constant over seasons with a nominal value of 1367 W/m$^2$. In this study, the constant was set to 1322 W/m$^2$ for the coldest case and 1422 W/m$^2$ for the hottest case.

#### 3.2.2. Pre-Processing

The pre-processing of the on-orbit thermal analysis was similar to that of the structural analysis. The mesh was generated as a 3-D tetrahedral type. The number of elements and nodes of Timon were 68,833 and 28,833, respectively, and those of Pumbaa were 96,727 and 36,713, respectively. Table 4 lists the thermal properties of the main materials. Given the contact conditions, heat transfer by conduction occurred among the contacted parts and was mainly related to the thermal conductivity and specific heat. Heat transfer by radiation occurred among all parts facing each other and was related to the emissivity and solar absorptivity.

**Table 4.** Thermal properties of the main CubeSat materials.

| Materials | Thermal-Physical Properties | | | Thermal-Optical Properties | |
|---|---|---|---|---|---|
| | Density (kg/m$^3$) | Conductivity (W/m·K) | Specific Heat (J/kg·K) | Emissivity ($\varepsilon$) | Solar Absorptivity ($\alpha$) |
| Al 6061 | 2711 | 154.3 | 896 | 0.07 | 0.35 |
| FR-4 | 1850 | 0.3 | 1300 | 0.6 | 0.6 |
| SS 304 | 7900 | 16.3 | 500 | 0.15 | 0.5 |
| Copper | 8920 | 387 | 385 | 0.03 | 0.3 |

The initial temperature was set to 15 °C, and the number of orbital cycles was set to 15, considering a low earth orbit. The average heat dissipation by internal electrical components was estimated by multiplying the maximum power consumption, duty rate of operations, and conversion efficiency. To consider heat generation due to the power consumption of components in more detail, an additional electrical test is required. However, in this study, the electrical test was omitted, and conversion efficiency values obtained from other studies were used for the worst-case scenario to consider various uncertainties such as non-uniform heat dissipation from the components. Numerous studies assume that all consumed power can be converted to thermal energy with a conversion efficiency of 100%

as worst case [20–22]. Furthermore, several studies assume the conversion efficiency of less than 90% [23–25]. In this study, the conversion efficiency of 80% was selected given practical limitations from irregular power consumptions with subsystems' operational modes and uncertainty of heat dissipation of harness. Active thermal control was applied to the heaters on the LiPo battery and the PPS. Table 5 presents the maximum power consumption of the electrical components, the battery heater, and the PPS tank heater.

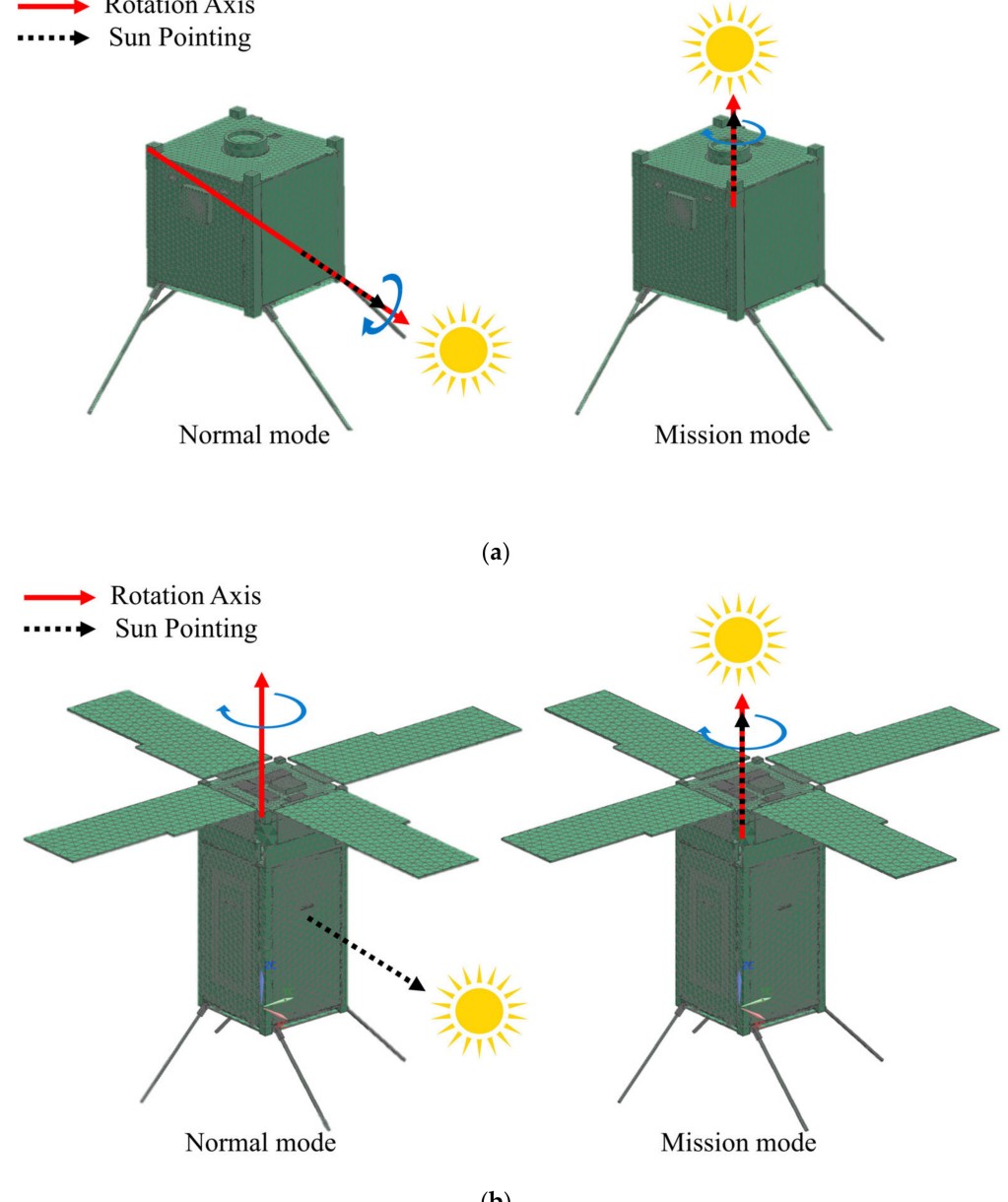

(**a**)

(**b**)

**Figure 9.** Attitude of CubeSats on the normal and mission mode: (**a**) Timon; (**b**) Pumbaa.

### 3.2.3. Analysis Results

Table 6 presents the results of the on-orbit thermal analysis with the operating and survival temperature ranges. Unlike the battery assembly which has operating temperature of $-10$–$50\,^{\circ}$C, the operating temperature of the battery cell depends on the condition. It is 0–45 $^{\circ}$C when the battery is charging, and $-20$–$60\,^{\circ}$C when the battery is discharging. The results were evaluated as the arithmetic average of all node temperatures in each component. In general, the temperature distribution of most components was not completely uniform; however, the difference between the maximum temperature and average

temperature was within 3 °C. Considering the temperature margin of 5 °C, we assumed a uniform temperature distribution and used the arithmetic average temperature.

**Table 5.** Power consumption of electrical components and active heaters.

| Hear Sources | Components | Maximum Power Consumption (W) | |
| --- | --- | --- | --- |
| | | 1U | 2U |
| Operation of electrical components | PCDU | 0.20 | |
| | Battery | 0.10 | |
| | MTQ | 0.75 | |
| | GNSS receiver | 1.50 | |
| | UHF transceiver | 0.25 | |
| | OBC | 0.50 | 0.70 |
| | RWA | N/A | 1.80 |
| | PPS | N/A | 0.20 |
| Active thermal control | Battery heater | 0.40 | 0.60 |
| | PPS tank heater | N/A | 1.5 |

Figure 10 shows the temperature profiles of the components on each CubeSat in the normal mode, where the margin of components except the visible camera exceeds 5 °C during operations. During the mission mode, the temperature was lower than that during the normal mode. As a result, the heaters of the LiPo battery and PPS were activated.

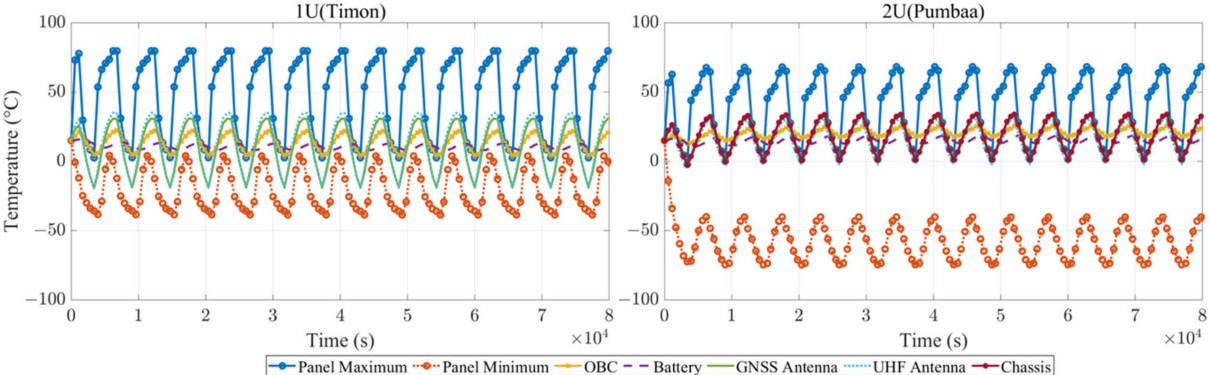

**Figure 10.** Thermal analysis results (normal mode).

Figure 11 shows the temperature profile of the Timon visible camera in the mission mode. To obtain images of the solar corona, the camera was activated on the dayside, and the temperature of the camera board was always higher than 0 °C within the operating temperature. The heaters of the LiPo battery were activated when the temperature was below 1 °C and deactivated when the temperature was above 6 °C. Figure 11 also shows the temperature of the LiPo battery with the heaters' activation cycle for the hottest case, where the heaters of both CubeSats are activated only in the mission mode, retaining a margin of 5 °C even in the coldest case.

**Table 6.** Survival temperature of main components and thermal analysis results.

| Subsystem | Components | Survival Temperature (°C) | | | | Analysis Results (°C) | | | | | | | |
|---|---|---|---|---|---|---|---|---|---|---|---|---|---|
| | | Operating | | Storage | | Timon (1U) | | | | Pumbaa (2U) | | | |
| | | | | | | Normal Mode | | Mission Mode | | Normal Mode | | Mission Mode | |
| | | Min. | Max. | Min. | Max. | Min. | Max. | Min. | Max. | Min. | Max. | Min. | Max. |
| PAY (Payload) | Visible camera | 0 | +60 | −40 | +85 | +3.3 | +17.1 | −0.4 | +10.9 | N/A | | N/A | |
| | Occulter board | −40 | +85 | −40 | +85 | N/A | | N/A | | −0.8 | +34.1 | −5.6 | +27.6 |
| CNDH (Command and Data Handling) | OBC | −40 | +85 | −40 | +85 | +1.5 | +21.9 | −8.8 | +3.0 | +14.3 | +25.5 | +2.2 | +7.3 |
| AOCS (Attitude and Orbit Control) | MTQ | −40 | +70 | −40 | +85 | −19.1 | +31.1 | −24.1 | +12.7 | −22.7 | +55.6 | −31.8 | +7.8 |
| | Sun sensor | −40 | +85 | −40 | +85 | −18.0 | +31.1 | −24.0 | +12.8 | −0.9 | +33.6 | −5.6 | +29.4 |
| | RWA | −20 | +60 | −20 | +85 | N/A | | N/A | | +6.7 | +26.9 | −4.1 | +6.7 |
| | GNSS antenna | −40 | +85 | −40 | +85 | −18.0 | +30.9 | −24.0 | +12.6 | −0.9 | +33.8 | −5.6 | +29.2 |
| | GNSS receiver | −40 | +85 | −55 | +95 | +9.1 | +16.0 | −6.5 | −3.1 | +15.8 | +23.8 | +3.7 | +7.5 |
| | PPS tank | 0 | +50 | −40 | +85 | N/A | | N/A | | +12.8 | +23.0 | +10.9 | +16.1 |
| COMS (Communications) | UHF antenna | −40 | +85 | −40 | +85 | −18.0 | +35.0 | −25.4 | +11.6 | −4.7 | +30.4 | −13.9 | +12.4 |
| | UHF transceiver | −30 | +85 | −30 | +85 | +10.2 | +18.9 | −5.6 | −1.7 | +15.7 | +24.2 | +4.0 | +7.9 |
| EPS (Electrical Power) | PCDU | −40 | +85 | −50 | +100 | −1.8 | +18.7 | −7.5 | +6.0 | −1.3 | +23.5 | −12.5 | +5.6 |
| | Battery Assembly | −10 | +50 | −20 | +60 | +5.1 | +12.8 | +1.2 | +5.8 | +9.1 | +18.7 | +0.7 | +5.7 |
| | Solar array | −100 | +150 | N/A | | −40.0 | +79.8 | −65.4 | +87.1 | −74.5 | +68.3 | −57.2 | +87.6 |
| SMS (Structure and Mechanism) | Chassis | N/A | | N/A | | −19.1 | +31.1 | −24.0 | +12.7 | −1.6 | +34.0 | −7.1 | +21.9 |

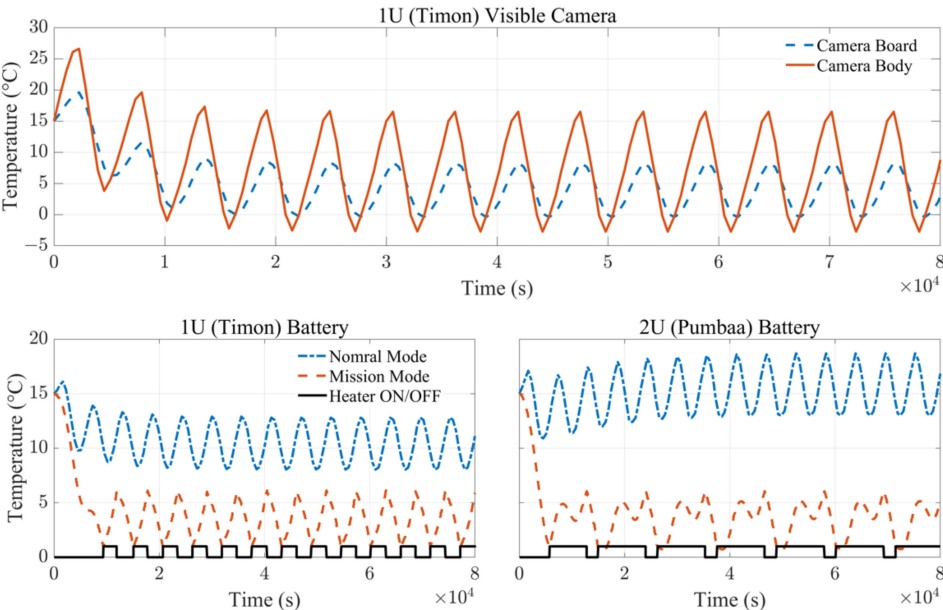

**Figure 11.** Temperature of visible camera and battery and operation of battery heater.

## 4. Ground Validations

Vibration tests and TVCTs were conducted to validate the structural-thermal design on the ground. Following the structural analysis results, the $f_0$ and structural responses were measured through vibration tests. Using TVCTs, the functionality of the CubeSats, such as algorithm computation and heater activation, was tested in the worst hot and cold case.

### 4.1. Vibration Testing; Launch Environment

For the vibration testing, five accelerometers were attached to the exterior of the CubeSats, as shown in Figure 12: the shaker fixture as a control reference, the test P-POD (T-POD), and several points on Pumbaa. Two types of shaker were utilized for the testing. Table 7 lists the specifications of the accelerometer and the two types of shaker. Figure 13 shows the test setup with the T-POD containing the CubeSats. Figure 14 shows the vibration-testing procedure. To evaluate the physical damage to the CubeSats, the low-level sine sweep (LLSS) test was conducted between the main tests, the quasi-static acceleration, and three stages of the random vibration. The sine burst test, which simulates the quasi-static acceleration, is executed over 10 cycles with a magnitude of 10G in each axis. The random vibration tests consisted of three stages: 7.42, 3.58, and 2.59 $G_{RMS}$, respectively. Between each test, a visual inspection and functionality test were carried out to evaluate the mechanical and electrical damages, respectively.

**Table 7.** Vibration simulator and accelerometer specification.

| Vibration Simulator | | | Accelerometer | |
|---|---|---|---|---|
| **Model** | **STI DC-8000** | **LDS -V984** | **Model** | **PCB 356B21** |
| Frequency range (Hz) | 5–2500 | 5–2000 | Frequency range (Hz) | 2–10,000 |
| Sine force (kN) | 78.4 | 160.1 | Sensitivity (mV/G) | 10 |
| Random force ($kN_{RMS}$) | 78.4 | 160.1 | Measurement range ($G_{pk}$) | ±500 |
| Maximum acceleration (m/s$^2$) | 980 | 981 | Resonant frequency (kHz) | ≥55 |
| Maximum velocity (m/s) | 2.0 | 1.5 | Broadband resolution ($G_{RMS}$) | 0.004 |

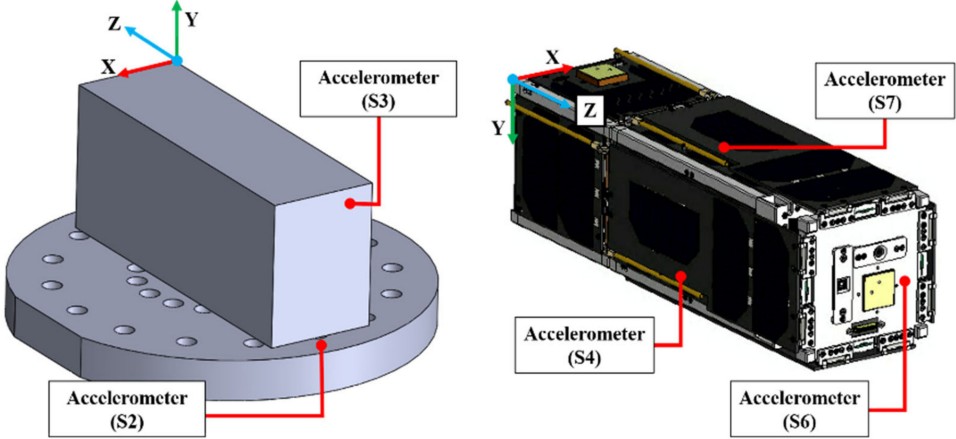

**Figure 12.** Accelerometer attachment locations. The three accelerometers were located in Pumbaa's −X-axis DSP, −Y-axis DSP, and +Z-axis panel. The other two accelerometers were located in the test P-POD and fixture.

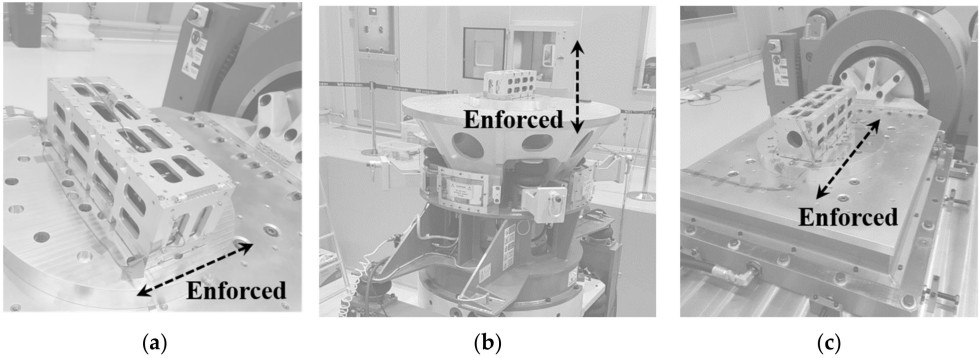

(**a**)       (**b**)       (**c**)

**Figure 13.** Vibration testing configuration: (**a**) X-axis test; (**b**) Y-axis test; (**c**) Z-axis test.

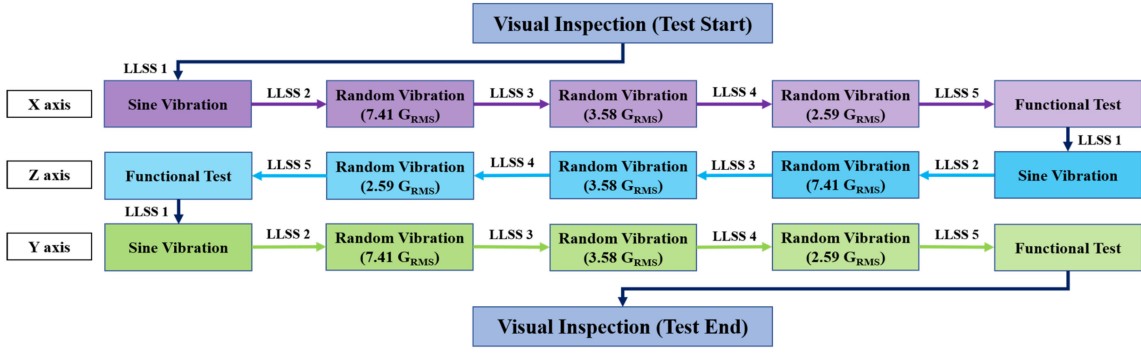

**Figure 14.** Vibration testing flow diagram.

At the final LLSS test, the $f_0$ in the axial and lateral directions was measured to be 153.2 and 74.3 Hz, respectively. The differences in $f_0$ measured in the LLSS test and numerical analysis were induced by a loose nylon wire of the HRM on the DSP. As there was no mechanical or electrical damage to either CubeSats over the entire vibration testing, it is concluded that the CubeSats will be safe within the launch environment.

## 4.2. Thermal Vacuum and Cycling Test: Space Environment

The TVCT with the thermal vacuum chamber simulated extreme temperature changes and high vacuum conditions. Figure 15a shows both CubeSats with thermocouples (TCs) in the thermal vacuum chamber. The CubeSats were fixed on the base plate using me-

chanical supports, which made them thermally isolated by conduction heat transfer from the base plate. Five and eight TCs were attached to Timon and Pumbaa, as shown in Figure 15b,c, respectively. The DSP in the −X-axis was deployed to attach a TC to the body-mounted panel.

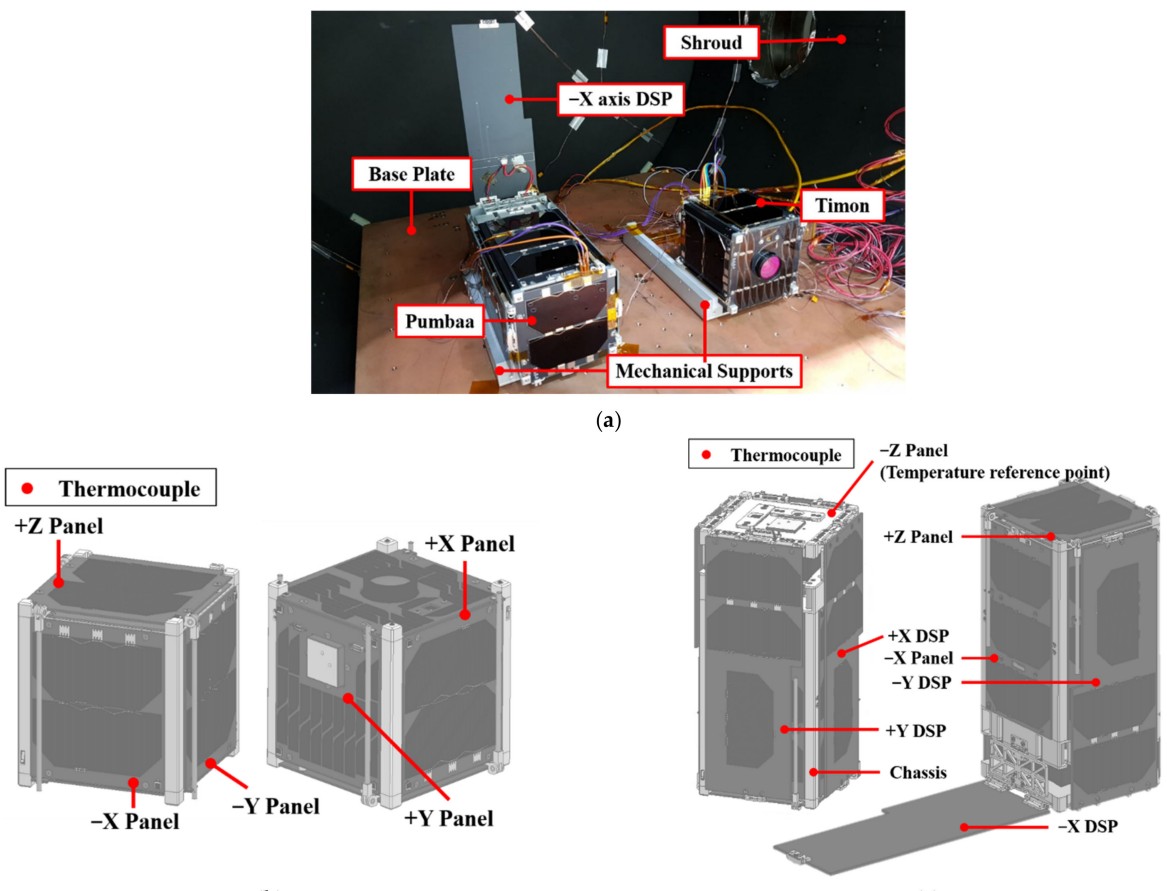

**Figure 15.** TVCT setting: (**a**) CubeSat arrangement in the chamber; (**b**) TCs on Timon; (**c**) TCs on Pumbaa.

The temperature profile of the TVCT is controlled by the temperature reference point (TRP) attached to a thermally stable point [26]. During the TVCT, the Z-axis aluminum panel on Pumbaa was set to the TRP following the results of the on-orbit thermal analysis. The temperature range of the TVCT was set from −15 °C to +30 °C, and the vacuum pressure was set lower than $4.0 \times 10^{-7}$ bar. Figure 16 shows the TVCT procedure, which involves 2.5 cycles and a dwell time of 3 h. During the dwell time, functional tests were carried out, including the operations of each component, formation flying algorithm computation, and firing of the PPS; in the cold case, the heaters on the PPS were evaluated by varying the duty rate. The total mass loss (TML) of the CubeSats was measured to be 4 g, and the TML was 0.36% of the total mass of Timon and was 0.16% of the total mass of Pumbaa. These values are below the TML requirement of 1% to prevent contamination in a high-vacuum space environment [27].

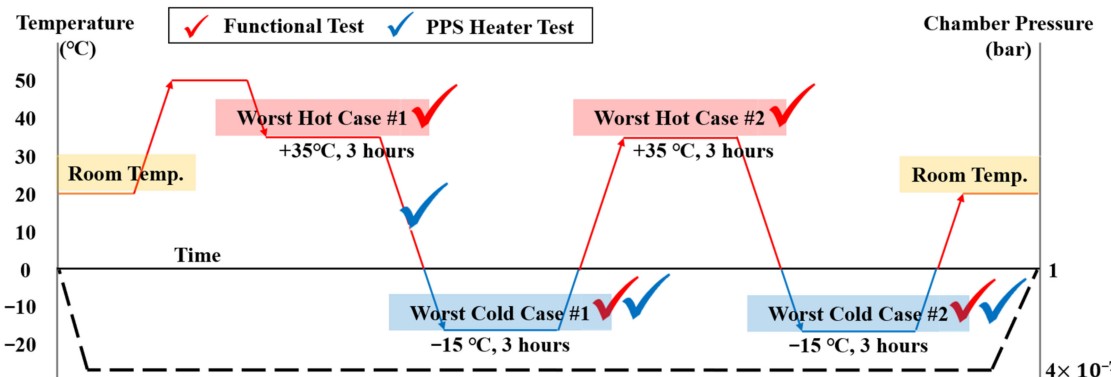

**Figure 16.** Temperature profile and procedure of TVCT.

## 5. Discussions: Environment Testing Results and Correlations

### 5.1. Vibration Testing

Table 8 and Figure 17 summarize the changes in $f_0$ in each axis over the vibration tests obtained from the accelerometers located in Pumbaa. It can be seen that in the first and final LLSS tests, the $f_0$ measured by the accelerometers was higher than 50 Hz for all axes, which is sufficiently high to avoid resonance with the deployer and launch vehicle. The values of the $f_0$ measured on the S4 and S7 accelerometers varied irregularly during the test. From Figure 18, it can be seen that the $f_0$ measured on S4, which was located in the X-axis DSP of Pumbaa, changed rapidly during the test. The values of $f_0$ change by approximately 2% in the X- and Z-axis.

**Table 8.** Summary of the natural frequency on each axis of the Pumbaa.

| | Axis | X | Y | Z |
|---|---|---|---|---|
| 1st mode natural frequency | Numerical analysis | 122.0 Hz | 120.6 Hz | 120.6 Hz |
| | First LLSS test | 105.5 Hz | 91.9 Hz | 156.6 Hz |
| | Final LLSS test | 108.1 Hz | 74.3 Hz | 153.2 Hz |
| Natural frequency change | Numerical analysis/First LLSS test | −13.5% | −23.8% | +29.9% |
| | First/Final LLSS test | +2.5% | −19.2% | −2.2% |

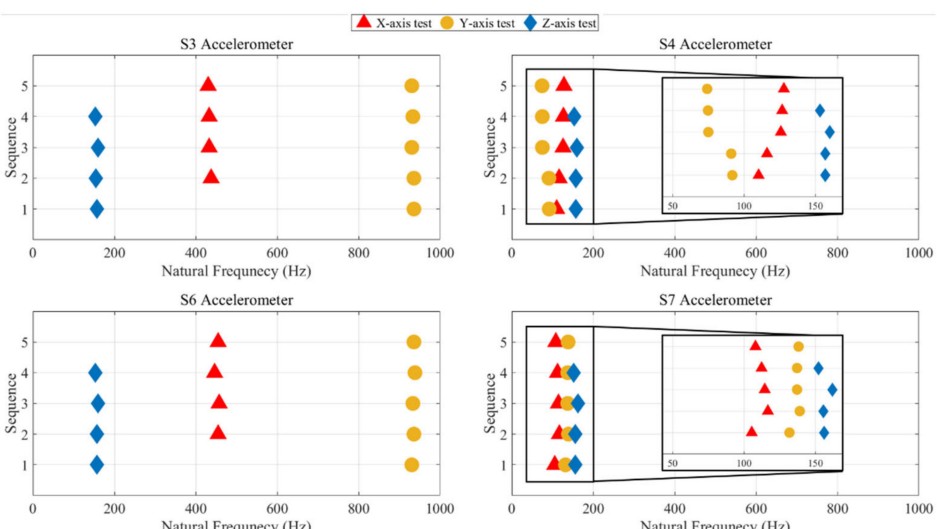

**Figure 17.** Modal survey results of the natural frequencies. The S3, S4, S6, and S7 accelerometers are located in the T-POD, −X-axis DSP of Pumbaa, −Y-axis DSP of Pumbaa, and +Z-axis panel of Pumbaa, respectively.

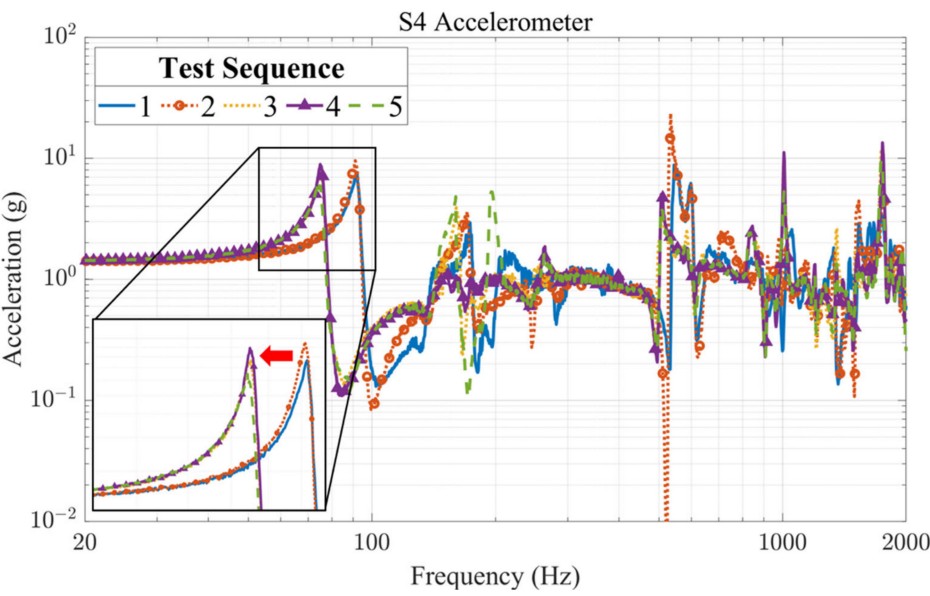

**Figure 18.** Change in natural frequency along the X-axis LLSS test as test sequence.

The HRM with a nylon wire cutting mechanism, which is mostly adopted for nanosatellites, has a limitation in that the natural frequency is unstable [6]. Considering that changes in the $f_0$ caused by the HRM are inevitable, the change in the $f_0$ of 6% measured on the X- and Z-axes is allowable change. However, in the Y-axis test, the $f_0$ of the DSP along the X-axis was found to decrease by 19.2% from the first LLSS test to the final LLSS test, which is a relatively large decrease compared to those found in other studies [28]. During the test for the Y-axis, the nylon wire of the HRM on the DSP was loosened, and the test was re-executed after repair, inducing a sudden change in $f_0$, as shown in Figure 18.

The $f_0$ measured in the X- and Y-axis tests was lower than the results of the structural analysis, but that in the Z-axis test was higher than the results of the structural analysis. The constraints of the backlash in the torsional hinge and tightening levels of the nylon wire were not implemented sufficiently in the numerical analysis [28]. To enhance the reliability of the analysis results, the constraint for tightening the nylon wire was considered during pre-processing and verified through the FEA.

Irregular workmanship can be replicated by simply varying the tension in the wire. However, it was impossible to vary the tension in the wire during numerical analysis due to limitations of the commercial software. To overcome these limitations, different types of tightening level and HRM methods were applied in the numerical analysis to determine whether a change in tightening level leads to a similar $f_0$ shift in the vibration test. The analytical model and its constraints for the structural analysis were improved by tightening the nylon wire around the HRM and connecting the parts of the HRM in 1-D. Figure 19 shows schematics of constraints of nylon wire tightening under the three different conditions, namely strongly bounded, face-to-edge connection, and edge-to-edge connection. Figure 20 shows the responses on the X-axis DSP under the three conditions. With the strongly bounded condition, the two parts of the HRM are in direct contact, and the value of $f_0$ is 125.0 Hz. When the parts are indirectly connected by a nylon wire with 284 elements and 144 nodes involved, $f_0$ decreases to 108.5 Hz. In particular, for the edge-to-edge connections with 124 elements and 62 nodes involved, the value of $f_0$ was the lowest 88.9 Hz. Therefore, noticeable changes in $f_0$ during the vibration testing and differences between numerical analysis and test occur due to different tightening levels of nylon wire and irregular workmanship. Finally, based on the results of the vibration tests, the nonlinearity of the DSP was studied by varying the tightening levels in the structural analytical model of the nylon wire to ensure accurate numerical analysis. Because the change in $f_0$ due to the nylon wire of the HRM can be estimated using numerical analysis, part-level vibration tests of HRM are not necessary.

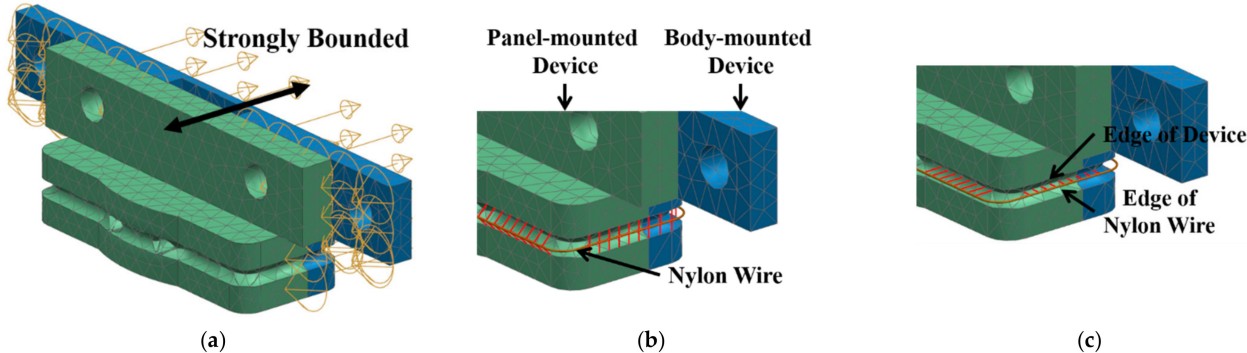

**Figure 19.** Three types of nylon wire tightening in the HRM of the Pumbaa: (**a**) Strongly bounded; (**b**) Nylon wire connected to edge of device; (**c**) Edge of nylon wire connected to edge of device.

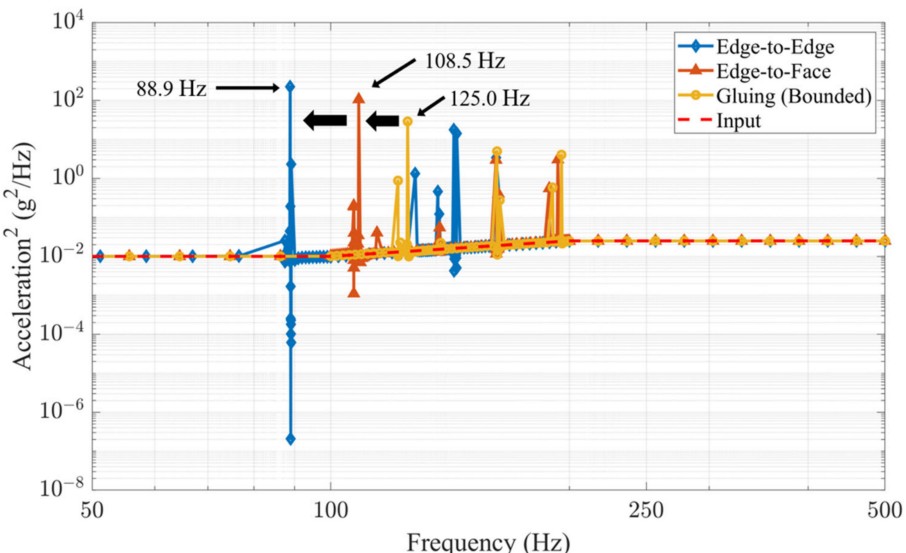

**Figure 20.** 1st mode natural frequency shifts as constraint level changes.

### 5.2. Thermal Vacuum and Cycling Test

Table 9 and Figure 21 show the temperature data from the embedded sensors and TCs during the TVCT of the two CubeSats. During the TCVT, the temperature of components and the chamber increased more than expected during the first hot dwell time due to an incorrect setting of the TRP at the chassis. Unlike in the numerical simulations, the rear side of the chassis was thermally shielded by Kapton tapes in the TCVT. During the first hot dwell time, the temperature of the components on Timon was higher than that on Pumbaa; the temperature of the OBC on Timon rose to approximately 70 °C, which is 20 °C higher than that of the OBC on Pumbaa. The temperature of the LiPo battery on Timon rose to approximately 63.5 °C, exceeding the operating temperature by 3.5 °C. After correcting the TRP setting from that of the chassis to that of the Z-axis aluminum panel on Pumbaa, the temperature of Timon's battery rose to 52.5 °C, which was 11 °C below the results of the first hot dwell time and had a margin of 7.5 °C from the operating temperature range. According to the certificate of conformance (CoC) document provided by the manufacturer, AAC Clyde Space, the calibrated coefficient for converting the analog value of the battery's built-in sensors to a digital value contains uncertainty of approximately 15% with respect to the theoretical value [29,30]. Although it is hard to ensure thermal safety from the TVCT temperature data, the battery operated effectively during functional tests before and after the TVCT. According to the aforementioned situations where the operational temperature range was exceeded only in the first hot dwell time, it was concluded that the TVCT had been passed.

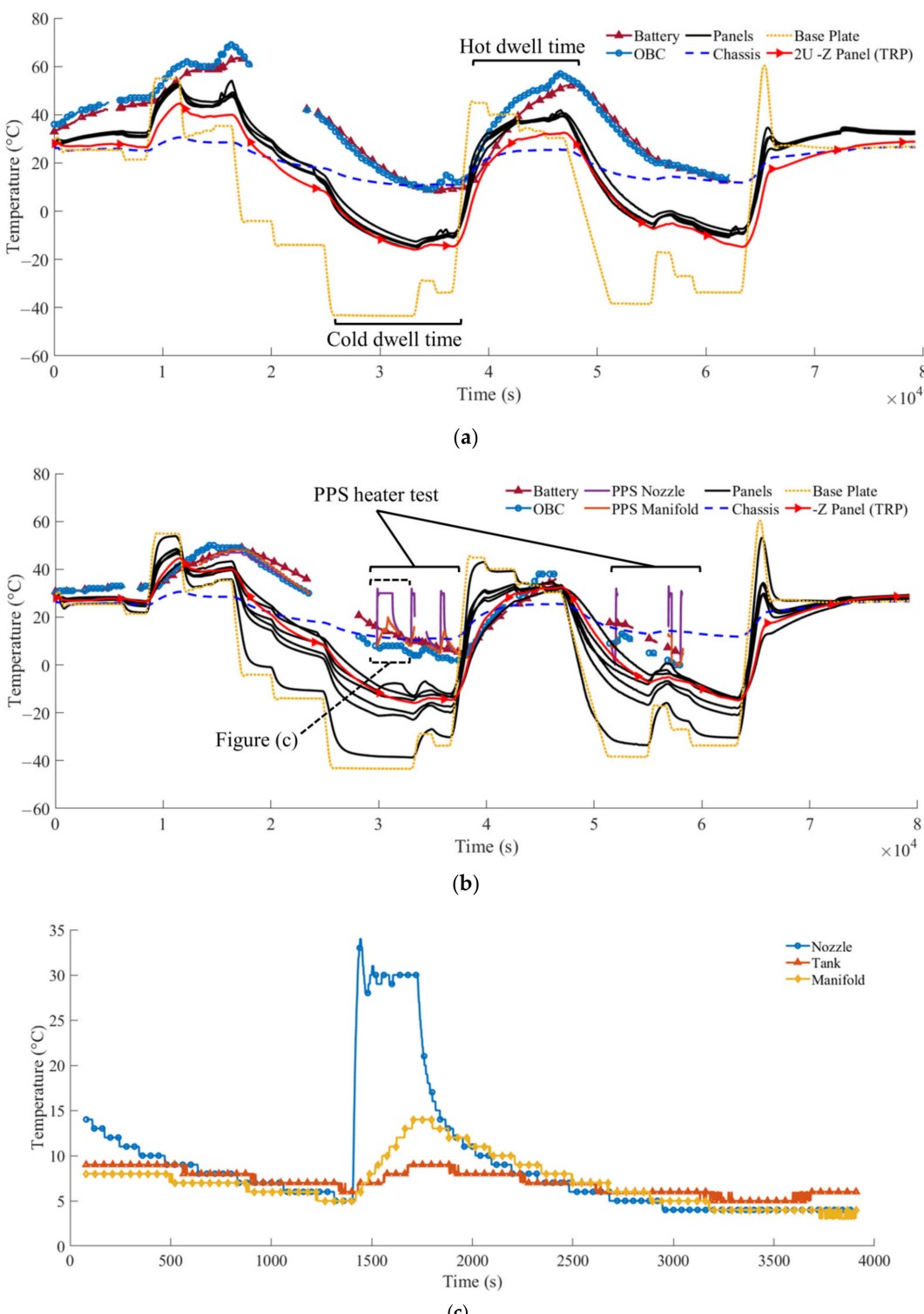

**Figure 21.** TVCT results: (**a**) Timon temperature profile; (**b**) Pumbaa temperature profile; (**c**) PPS temperature profile of Pumbaa during heater activation.

Figure 21c shows the temperature profiles of the PPS of Pumbaa. From the figure, it can be seen that when the heater on the nozzles and tank of the PPS were activated, the temperature rose up to the operating temperature within 30 s. The duty rate of the heaters was calibrated based on the results; it was set to a magnitude of 50% and a duration of 5 min as a default. The results verify that the heaters can successfully drive the PPS for orbit maneuvers even during eclipses.

**Table 9.** TVCT results.

| Components | | Temperature (°C) | | | |
|---|---|---|---|---|---|
| | | 'Timon | | Pumbaa | |
| | | Min. | Max. | Min. | Max. |
| Panel | −X | −11.0 | 50.0 | −9.0 | 47.0 |
| | +X | −13.0 | 55.0 | −4.0 | 46.0 |
| | +Z | −14.0 | 50.0 | −12.0 | 49.0 |
| DSP | −X | N/A | N/A | −38.7 | 53.9 |
| | −Y | N/A | N/A | −22.9 | 48.5 |
| Battery | | 8.0 | 63.5 | 4.3 | 49.4 |
| PCDU | | 10.6 | 65.8 | 4.7 | 52.0 |
| OBC | | 9.0 | 70.0 | 0 | 50.0 |
| RWA | | N/A | N/A | 12.5 | 51.3 |
| PPS | | N/A | N/A | 0 | 47.0 |
| Chassis | | N/A | N/A | 10.4 | 30.6 |

For the TVCT, the temperature of the chassis changed less than that of the internal components, which is inconsistent with the thermal analysis results. Additionally, the temperature of each internal component was higher than that of the solar panels. To enhance the accuracy of the thermal analysis and consider the interactions between the CubeSats in the thermal vacuum chamber, the thermal analytical model and the heat transfer conditions were modified by duplicating the TVCT setting as shown in Figure 22. The duration of thermal cycling has a significant effect on the convergence temperature of the internal and external components of the CubeSats. Thus, for reliable correlation, the time required for the temperature of the TRP to change between the hot section and cold section was set to the same as that set for the TVCT, i.e., approximately 20,000 s.

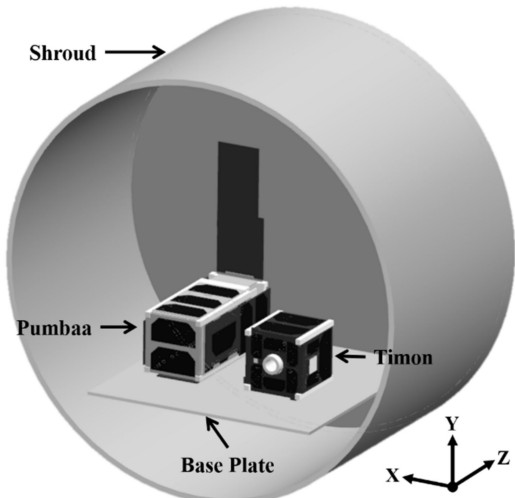

**Figure 22.** Duplicated thermal vacuum chamber and the CubeSats for the correlation of the TVCT parameters.

The thermal properties of the analytical models, including the heat dissipation and thermal conductance, were correlated using the TVCT data and numerical analysis for minimizing the temperature difference (ΔT). First, we correlated the heat dissipation of electrical components. To simulate the maximum heat dissipation of the real case, the duty cycle of 100% was considered. Heat dissipation of most components was higher than the predicted value calculated based on an efficiency of 80%. For instance, the heat dissipation of the PCDU was revised from the assumed value of 0.16 W in the initial on-orbit thermal analysis to the correlated value of 0.25 W after the TVCT. Second, we correlated the coupling condition between the occulter and chassis. Before the correlation,

because the occulter and chassis were connected to each other, the ΔT between the chassis, occulter, and TRP was small. By decreasing the conductance between the chassis and occulter, the TRP and the chassis were thermally separated, as in the TVCT. Lastly, the thermal-physical properties of the UHF transceiver and GNSS receiver were adjusted.

Figure 23 shows the ΔT between the TVCT results and the numerical analysis before and after the correlation. After the correlations, ΔT became smaller than 3 °C for at least 90% of the correlated components, ensuring the reliability of the thermal analytical model. The correlated model enhanced the reliability of the general thermal analysis. In addition, the required duty rate of the heaters can be predicted based on the reliable thermal analytical model for the power budget analysis of the systems.

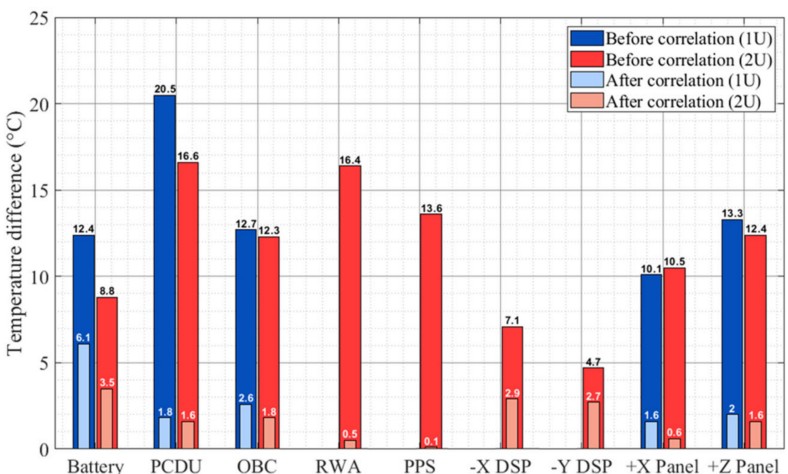

**Figure 23.** Temperature difference between the TVCT and the numerical analysis before and after the correlation.

With the correlated thermal analytical model, a numerical analysis was conducted without heat dissipation to analyze the cause of ΔT between the CubeSats in the TVCT. The ΔT between the CubeSats seems to be induced by structural characteristics, because the range of the temperature change of Timon is greater than that of Pumbaa without any heat dissipation. The structural characteristics that caused ΔT are as follows. First, the body of Pumbaa did not receive radiant heat directly because it was obscured by the DSP. On the other hand, Timon, which received radiant heat directly from all directions of the body, had a higher internal temperature than Pumbaa. Second, Timon has lower thermal mass and more efficient conduction paths than Pumbaa. Timon is relatively small, with a lower thermal mass than Pumbaa, and is more sensitive to external radiation heat transfer because it has efficient conduction paths due to its simpler structure. For example, the board supporting rod on Timon is a single part, whereas the board supporting rod on Pumbaa consists of several parts, as shown in Figure 24. Meanwhile, the assembly, an accumulation of mechanical tolerances, makes the contact area imperfect. Specially, for the multiple-parts rod, the contact area between each rod could not be established due to a mechanical tolerance among boards and supports. This separation or partially contacted area lead the heat transfer by conduction is inefficient by increasing thermal resistances compared to the single-part rod. Therefore, owing to its efficient conduction paths and low thermal mass, Timon exhibits a wider range of temperature changes than Pumbaa. The temperature difference between the results of the analysis and the TVCT is discussed. The correlation scheme enables the reduction of the development cost of missions with multiple CubeSats by conducting TVCT of multiple CubeSats simultaneously.

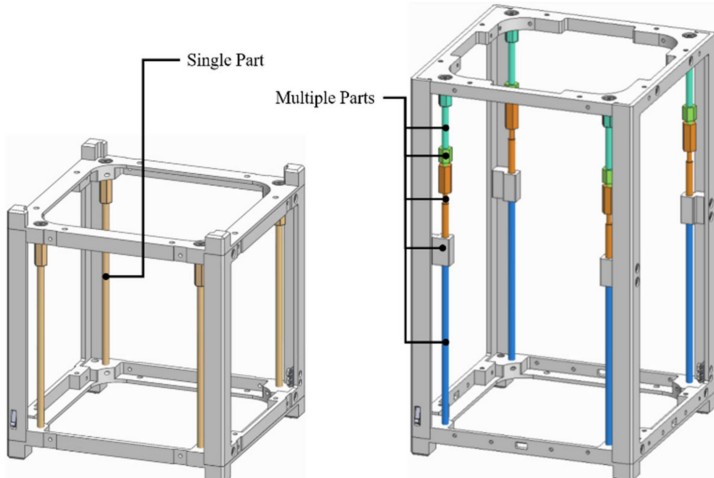

**Figure 24.** Configuration of board supporting rods of CubeSats (left: Timon, right: Pumbaa).

## 6. Conclusions

We developed a structural-thermal design of 1U and 2U CubeSat to ensure safety in the launch and space environments. The structural and thermal design was executed based on numerical analysis using the FEA. To improve the reliability of the numerical analysis, the nonlinearity of the DSP was studied by implementing the analytical model of a nylon wire. In addition, constraints of coupling multiple CubeSats in a single deployer were implemented. To validate the design, environmental tests, including vibration tests and TVCTs, were carried out. Vibration tests were carried out to check the structural safety of the acceleration and random vibrations in the launch environment. Although neither CubeSats experienced structural or functional damage during the test, the $f_0$ on the DSP decreased owing to the different tightening levels of the nylon wire and irregular workmanship. The change in $f_0$ of the DSP during vibration tests was implemented by applying different tightening levels to the nylon wire in the numerical analysis. Thus, a reliable structural analytical model that included an HRM with a nylon wire was developed. The thermal analytical models were correlated by comparing the results of TVCTs and numerical analyses. Furthermore, the structural characteristics of CubeSats with different sizes, which induced a temperature difference, were studied. To save time and costs required when developing multiple CubeSats, the TVCT was carried out simultaneously for the two CubeSats in the same chamber. In the functional test performed during the dwell time, all electrical components were operated normally, including an active heater of the micro-PPS. Based on the TVCT results, the thermal analytical model was correlated by implementing a thermal vacuum chamber in the numerical analysis. The thermal properties and heat dissipation of the electric components were adjusted to minimize $\Delta T$. Numerical analysis results based on the correlated analytical model ensured structural safety under the launch environment. According to the TVCT and on-orbit thermal analysis, it was concluded that all components met the thermal margin. Therefore, further studies relevant to the battery are required to ensure its thermal safety in the space environment.

**Author Contributions:** Conceptualization, Y.-K.P., G.-N.K. and S.-Y.P.; methodology, Y.-K.P., G.-N.K.; software, Y.-K.P.; validation, Y.-K.P., G.-N.K. and S.-Y.P.; formal analysis, Y.-K.P., G.-N.K. and S.-Y.P.; writing—original draft preparation, Y.-K.P.; writing—review and editing, G.-N.K. and S.-Y.P.; supervision, S.-Y.P.; project administration, G.-N.K. and S.-Y.P.; funding acquisition, S.-Y.P. All authors have read and agreed to the published version of the manuscript.

**Funding:** This work was funded by the Ministry of Science and ICT of the Republic of Korea.

**Institutional Review Board Statement:** Not applicable.

**Informed Consent Statement:** Not applicable.

**Data Availability Statement:** The data used to support the findings of this study are available from the corresponding author upon request.

**Acknowledgments:** This work was supported by the Space Basic Technology Development Program through the National Research Foundation (NRF) of Korea, funded by the Ministry of Science and ICT of the Republic of Korea [grant number NRF-2017M1A3A3A06085349].

**Conflicts of Interest:** The authors declare no conflict of interest.

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
