# Peer review of "Novel Structure and Thermal Design and Analysis for CubeSats in Formation Flying"

_aerospace, doi:10.3390/aerospace8060150_

Round 1
Reviewer 1 Report
This paper presents the numerical and experimental verification of the two CubeSat composing the CANYVAL-C mission. Both spacecraft have been modelled for structural and thermal analysis, with enough detail and satisfying results. Sine and random vibration tests were performed employing the launch loads provided by the selected launcher datasheet. Discrepancies in the first natural frequencies were correlated to the nylon wire tightening system for solar panels deployment; numerical modelling was employed to reproduce this behaviour and no further vibration experiments were deemed necessary. With respect to the thermo-vacuum tests, minor discrepancies between the detailed thermal model and experiments were justified by a set of simulations replicating the experimental setup; the thermal model was consequently updated.
In general, the paper is well written and well organized. It addresses the most important points, with enough details in describing the simulations and the experiments strategies.
The following minor suggestions are given to improve the quality of the manuscript.
- A picture and a reference to the CANYVAL-C mission are reported, but it would be useful to summarize it with a few more sentences in section 1.
- Page 2, lines 69-70. Please explain this sentence. It is not clear how including the HRM would meet the requirements of power generation.
- Page 5, line 247. If possible, please justify the choice of a design safety factor of 2.
- Page 7, line 195. Please fix the typo that was exported in pdf as a large space.
- Page 7, lines 198-202. To improve readability, please introduce that T is the initial thickness.
- Page 11, line 267. Please check the elements and nodes number, as Timon has 68,905 elements with 68,148 nodes and Pumbaa 96,727 elements with only 36,713 nodes, about 1/3rd.
- Page 16, line 362. Please use bar or Pa instead of torr as pressure unit.
- In general, during or after the experimental vibration campaign a shock test is performed. Please discuss it and define if a shock test is planned for the future.
Author Response
Please see the attached file for author's responses. Thank you very much.

Reviewer 2 Report
Dear Editor,
The paper provides an interesting study concerning the structural and thermal designs of the CubeSats Timon and Pumbaa through finite element analyses. The work is well organized, written and structured, and the analysis is well conducted.
To the reviewer opinion, the quality of the work is suitable for publication as it is in the present form.
Author Response
We thank for your insightful comments and recommendations.
Reviewer 3 Report
Major Comments:
1) I fail to understand why the different levels of tightening in the HRM were studied by changing the type of bonding between the wire and the device. A more correct approach would be to simply vary the tension in the wire and study the vibrational response of the system. Also, the authors keep referring to an 'analytical model', especially in lines 416-418. I am not sure what they mean by an analytical model. I don't think an analytical model was presented in this paper. We only saw a computational/numerical model and experimental data.
2) Given that your experimental data showed that the temperature of the LiPo battery was 13C higher than the operating temperature range, I don't think you can justify your conclusion that your results are ensuring thermal safety of the satellite. In fact, just like a safety margin was included in the structural analysis, the authors are advised to include a safety margin in their operating temperature specifications and to ensure that the design meets those specifications.
3) The 'correlation' between the thermal model and experimental results sounded a lot more like 'fudging of parameters' which is okay in some situations as long as the goal is to get a phenomenological, useful model for engineering applications. But for a research paper, I would expect the authors to go deeper into the source of the discrepancy between their analytical model and experimental results. For instance, if the heat dissipation of PDCU was increased from 0.05 W to 0.25 W, this should have been accompanied by actual electrical tests of the PDCU to verify what the actual power dissipation was. There should in general be a justification for the 'adjustment' of various thermal parameters. On a related note, 80% efficiency was an unjustified assumption to begin with. I could not find a justification for that efficiency number in the reference [19] cited by the authors.
4) The justification given by the authors for the large temperature swings in Timon is not very convincing. Why would the presence of multiple parts necessarily lead to lower temperature swings? A more likely cause is simply the fact that Pumbaa is a 2U while Timon is a 1U CubeSat. Timon hence has a lower thermal mass and is likely to experience larger temperature swings since the effect of the external radiation is felt by all the internal components more readily than in the case of Pumbaa where the conduction paths may not be as efficient.
Minor Comments:
1) On line 58, the lack of convection is not due to microgravity but simply due to the vacuum environment
2) Missing text on Line 195
3) It is unclear what is being shown in the left image in Fig. 5(b)
4) Contrary to what is said in Line 281, Table 5 does not contain the activation conditions for the heaters. It is described later in the text of the manuscript.
5) I don't think it is okay to take the arithmetic average of all the nodes in a component to evaluate the thermal safety of the system. Some parts of the component could be at a much higher temperature leading to a failure of the component.
6) The authors never clearly described the Conops of mission mode and normal mode. For what fraction of an orbit is the satellite in mission mode and for what fraction is it in normal mode?
Round 2
Reviewer 3 Report
While the authors have addressed a majority of the comments, some comments are still not addressed properly in the new version of the manuscript.
1) The heat dissipation of PDCU was increased from 0.05 W to 0.25 W. This is a factor of 5 increase in power dissipation. This has to be accompanied by aa justification for the 'adjustment'.
2) The authors themselves state that assuming 100% conversion efficiency would be the worst case but they still persist with their 80% conversion efficiency assumption, which is not aptly justified in the manuscript.
3) The authors continue to insist that the conduction paths are less efficient in Pumbaa because of multiple parts. I don't see a justification for why multiple parts leads to less efficient conduction paths.
Round 3
Reviewer 3 Report
On Line 516, I don't understand how the value of 0.05 changed to 0.16. While the explanation provided by the authors is satisfactory, they must be consistent in the numbers presented.
